# One-Pot Environmentally Friendly Synthesis of Nanomaterials Based on Phytate-Coated Fe_3_O_4_ Nanoparticles for Efficient Removal of the Radioactive Metal Ions ^90^Sr, ^90^Y and (UO_2_)^2+^ from Water

**DOI:** 10.3390/nano12244383

**Published:** 2022-12-09

**Authors:** Paulino Duel, María de las Nieves Piña, Jeroni Morey

**Affiliations:** Departament de Química, Universitat de les Illes Balears, Cra. de Valldemossa km 7.5, 07122 Palma de Mallorca, Spain

**Keywords:** adsorption removal, radioactive metal ions, magnetite nanoparticles, phytate, one-pot synthesis, water contamination

## Abstract

We report the fast (three minutes) synthesis of green nanoparticles based on nanoparticles coated with the natural organic receptor phytate for the recognition and capture of ^90^Sr, ^90^Y, and (UO_2)_^2+^. The new material shows excellent retention for (UO_2_)^2+^, 97%; these values were 73% and 100% for ^90^Sr and ^90^Y, respectively. Recovery of the three radioactive metal ions occurs through a non-competitive process. The new hybrid material is harmless, easy to prepare, and immobilizes these radioactive contaminants in water with great efficiency.

## 1. Introduction

The use of nuclear energy, as well as the isolation and handling of radioisotopes in medicine and industry, is one of the greatest achievements of modern science and technology. However, nuclear energy also generates highly polluting and hazardous waste. The world has recently closed many nuclear power plants that generate tons of highly contaminated waste.

The radioactive lifetime of some radioisotopes can be very long. The half-life of uranium is 2.48 × 10^5^ years, and it is the largest waste generated by a nuclear power plant; uranium is a highly toxic and carcinogenic waste. The maximum level established by the EPA for the ambient concentration of uranium is 30 μg/L [1].

Yttrium has an abundance in the Earth’s crust of 0.0028% [2]. It is used industrially in the construction of lasers, and, together with neodymium, is used to remove tattoos. It is also used with aluminum for the generation of artificial diamonds [3]. The radioactive isotope ^90^Y is used for selective internal radiation therapy (SIRT) to treat liver cancers [4,5]. Yttrium is a toxic metal that can damage the eyes and lungs [6]. The maximum concentration allowed in water for human consumption is 6.4 μg/L [7].

Strontium is an essential element in the human body at trace levels. Strontium stimulates the creation of bone material and has been directly related to the compressive strength of bones. Strontium ranelate inhibits bone demineralization in patients with postmenopausal osteoporosis. However, an excess of this metal inhibits bone mineralization. The chemical similarity of Sr^2+^ with Ca^2+^ causes displacement of calcium from bones leading to malformations or dysfunctions in the skeleton [8]. Sr^2+^ absorption can also act with vitamin D to cause adverse effects on the synthesis of 1,25-dihydroxyvitamin D3, thus directly affecting the skeleton and causing rickets in animal models [9].

Strontium occurs in nature with different stable isotopes (^84^Sr, ^86^Sr, ^87^Sr, and ^88^Sr). Isotopes ^85^Sr, ^89^Sr, and ^90^Sr are naturally radioactive. ^90^Sr is the most dangerous radioactive isotope with a half-life of 28.79 years. This can cause prolonged beta emission with long exposure times. ^90^Sr decay generates radioisotope ^90^Y that subsequently decays to ^90^Zr as a stable isotope. The half-life of yttrium-90 is 64.1 h. It is a high-energy ß-emitter (E_max_ = 2.27 MeV) [10], and it emits beta radiation faster than ^90^Sr. Radioactive strontium is generated by nuclear fission in the explosion of nuclear weapons or during accidents in nuclear plants. It is generated mostly as ^90^Sr to generate particles with other components (<1 μm) that can travel worldwide depending on their size. The accumulation of ^90^Sr in bones leads to direct exposure of bone and muscle cells to beta radiation, thus causing DNA deterioration and generating cancer cells [11]. To summarize, U(VI), ^90^Sr, and ^90^Y are persistent and very toxic contaminants. Thus, their controlled storage is a key goal.

Hybrid magnetite nanoparticles have been proposed as an alternative for environmental remediation [12,13,14,15] including the removal of noble metal salts (Ag and Au [16]) or salts of heavy metals contaminants such as Hg(II) and Pb(II) [17]. They have also proven useful in the capture of polycyclic aromatic hydrocarbons (PAHs) [18]. More recently, hybrid magnetite nanoparticles are used in the quantitative determination of volatile organic compounds (VOCs) [19].

All this is the result of the low cost and the great versatility in the preparation of hybrid magnetite nanoparticles, which are coordinated with different organic remains and can join with different target molecules. The contaminant is generally dissolved or is present in water. The inherent magnetism of hybrid magnetite nanoparticles offers simple and effective removal of the contaminant from the sample by magneto filtration once coordinated on its surface. This obviates the need to use external filters. Magnetism also allows one to concentrate the complex formed by the association of hybrid magnetite nanoparticles with the analyte. This leads to greater control over the captured analyte and is a particularly promising solution to collect and store highly toxic and dangerous radioactive substances. One can then store and maintain them in a monitored environment.

The phosphonate group can form a strong bond with the surface of Fe_3_O_4_ nanoparticles (FeNPs). This process coats their surface and protects them from oxidation of Fe(II) to Fe(III) [20] and prevents their lixiviation. The phosphate group also effectively coordinates with the salts of uranium U(VI), Sr(II), and Y(III) [21,22,23,24,25].

Phytate is a natural and economical source of phosphate groups. Phytate is a natural product produced by plants and serves as a reserve of *myo*-inositol and phosphates for plants. Chemically, phytate is the dihydrogen phosphate ester of *myo*-inositol, or *cis*-1,2,3,5-trans-4,6-cyclohexanehexol. The most abundant and thermodynamically stable conformer has five groups of the six phosphates in equatorial positions (1a5e) [26]. However, the molecule can reverse the orientation from equatorial to axial (5a1e) at a pH between 9.0–9.5 [27,28].

Here, we present an efficient single-step synthesis of magnetite nanoparticles coated with phytate (Phy@Fe_3_O_4_). This method is simple and fast (<3 min) and does not require the use of organic solvents. Only water is required and there are no special reaction conditions (anhydrous reagents, inert atmosphere, or controlled temperature or catalysts). The Phy@Fe_3_O_4_ show excellent uranium uptake: approximately 950 milligrams of uranium per milligram of Phy@Fe_3_O_4_ in aqueous samples of 100 ppm uranium. These were made under normal environmental conditions and show a high affinity for salts of ^90^Y(III) and ^90^Sr(II) [29].

## 2. Materials and Methods

### 2.1. Reagents

All commercially available reagents were of analytical grade. Inositol hexaphosphoric acid sodium salt (phytic acid sodium salt hydrate) from rice, strontium standard for ICP *Trace*CERT^®^ HNO_3_ 2% *w*/*w*, yttrium standard for ICP *Trace*CERT^®^ HNO_3_ 2% *w*/*w*, and uranium ICP standard UO_2_(NO_3_)_2_·6H_2_O in HNO_3_ 2–3% 10 mg/L U were purchased from Sigma-Aldrich (Waltham, MA, USA). All solvents were purchased from Scharlau and Fisher Chemicals. Acros Organics supplied iron (II) chloride tetrahydrate (FeCl_2_·4H_2_O) and iron (III) chloride (FeCl_3_ anhydrous). High purity water was generated by Milli-Q apparatus (Millipore Corp., Madrid, Spain).

### 2.2. Characterization of the Nanomaterials

Reactions and the sorption experiments used oven-dried glassware. The nanomaterials were characterized with optical, thermal, spectroscopic, and microscopic techniques.

Infrared spectra (FT-IR) analysis of samples used a Bruker Tensor 27 instrument (Bruker Española, Madrid, Spain) by mixing dried Phy@Fe_3_O_4_ powders into KBr pellets. The spectra were collected with a resolution of 4 cm^−1^ from 4000–400 cm^−1^.

Thermogravimetric analysis (TGA) used a SDT Q600 TA (TA Instruments, New Castle, DE, USA) in Al_2_O_3_ crucibles. Dried samples (typically 7–8 mg) were initially held at 40 °C for 10 min before heating to 600 °C at a rate of 10 °C min^−1^ in an N_2_ atmosphere.

Inductively coupled plasma-optical emission spectrometry (ICP-OES) was performed on a Perkin-Elmer Optima 5300 DV instrument (Thermo Fisher Scientific, Waltham, MA, USA).

Transmission electron microscopy (TEM) used a Philips Jeol JEM 2100F (Jeol, Zurich, Switzerland). A drop of the different nanoparticle suspensions was drop-cast on a copper TEM grid with a carbowax backing.

Size and zeta potential were measured via dynamic light scattering (DLS) with a Malvern Zetasizer Nano ZS (Malvern Panalytical, Malvern, UK) at 25 °C.

A NdBFe neodymium magnet (60 mm × 30 mm × 15 mm and 549.4 M of maximum magnetic force) was from Supermagnete (Gottmadingen, Germany). A flatbed orbital shaker IKA VXR basic Vibrax was also used (IKA-Werke GmbH & Co. KG, Staufen, Germany).

Bremsstrahlung radiation from ^90^Y(III) was measured using a LB4200 Alpha-Beta radioisotope dose calibrator (Mirion Technologies, Canberra, Inc. Meriden, CT, USA). All measurements were performed under the same geometric conditions. The results are reported as the mean of at least three measurements ± standard deviation.

### 2.3. Quantification of Residual Metal Concentrations Using ICP-OES

The number of remaining cations in the solution after different mixing times were measured by ICP-OES. Here, an acid solution (HNO_3_ 2%) of all cations at a concentration of 1000 ppb was used. The ICP-OES was calibrated just before each measurement between 0 to 1000 ppb. The Phy@Fe_3_O_4_ were previously sonicated for 10–20 min to achieve a good dispersion.

In a 12 mL Falcon tube, 10 mL of a metal salt solution (1000 ppb) was introduced, followed by 0.5 mL of a suspension of functionalized phytate magnetic nanoparticles in milli-Q water. The amount of Phy@Fe_3_O_4_ depends on the concentration of the suspension and was precalculated. This mixture was stirred for two hours. The suspension was then decanted using an NdBFe magnet, and the supernatant was filtered with a 0.45-micrometer filter to confirm that no nanoparticles entered the instrument. The elimination efficiency and the adsorption capacity at equilibrium, *q_m_* (mg/g), were calculated as follows using the data obtained in these experiments (see Equations).
Removal (%) = [(*C_o_* − *C_t_*)/*C_o_*] × 100%
*q_m_* = (*C_o_* − *C_e_*) *V*/*m*

The *C_o_* (mol/L) is the initial concentration of metallic salt, *C_t_* (mol/L) is the concentration of metallic salt after adsorption at time *t*, and *C_e_* (mg/L) is the equilibrium concentration. *V* (L) is the volume of the solution used for the extraction experiment and *m* is the mass of nanoparticles used in the experiments, expressed in grams.

### 2.4. Quantification of Residual ^90^Sr/^90^Y Activity

A solution of ^90^Sr/^90^Y in equilibrium was used. The results initially showed a standard activity of 99.49 ± 0.2 Bq/g (15 October 1998). On the day of the experiment, 10 May 2021, the recorded activity was 57.88 Bq/g. Measurements were made using 4.84 g of this standard solution dissolved in 20 mL of milli-Q water at pH 6.5. The standard solution (20 mL) was divided into four 10-mL glass tubes. The total activity measured was 280.14 Bq.

Next, 2.5 mg of Phy@Fe_3_O_4_ were added to each tube. The samples were shaken in an orbital flatbed shaker for 2 h. The Phy@Fe_3_O_4_ were then collected from each aliquot using an NdBFe magnet. From each aliquot, 3 mL of the supernatant solution was extracted and poured onto an aluminum plate. The sample was dried in an oven at 120 °C, and the radioactive activity was then measured for each of the four dry wastes. This measure corresponds to the remaining activity of the solution.

Next, the activity of each aliquot of the decanted Phy@Fe_3_O_4_ nanoparticles was measured. The nanoparticles were dispersed with the remaining 2 mL of the solution and filtered with a 0.2-micron PVDF filter. The filter contained the Phy@Fe_3_O_4_ nanoparticles coated with ^90^Sr/^90^Y; they were dried in an oven set to 120 °C. The radioactivity of the four dry wastes was then measured separately. The measurement of the remaining activity indicates the retention capacity of the nanomaterial in each aliquot.

### 2.5. Synthesis of Fe_3_O_4_ and Phy@Fe_3_O_4_ Nanoparticles

#### 2.5.1. Synthesis of Fe_3_O_4_ Nanoparticles

A solution of NaOH 1 M (7 mL) was added to a stirred solution of FeCl_3_ (162 mg, 1 mmol) and FeCl_2_·4H_2_O (100 mg, 0.5 mmol) in Milli-Q water (5 mL) over 15–20 min under argon. When the addition was finished, the mixture was stirred for another 20 min. The black precipitate was collected by a NdBFe neodymium magnet and washed with Milli-Q water until the pH was neutral. Finally, nanoparticles were cleaned with methanol to remove water and suspended in 10 mL of methanol.

#### 2.5.2. Synthesis of Phy@Fe_3_O_4_ Nanoparticles

200 mg of FeCl_2_·4H_2_O (1.0 mmol), 320 mg of FeCl_3_ (2.0 mmol), and 400 mg of the phytic acid sodium salt (0.61 mmol) were dissolved in Milli-Q water (30 mL). A white precipitate was formed corresponding to the iron phytate salts. Next, 20 mL of NaOH (1 M) solution were added to the suspension, and the reaction was stirred for three minutes at 25 °C. The supernatant of the reaction was clear, and the black power was decanted using a NdBFe neodymium magnet. The product was cleaned with Milli-Q water (3 × 15 mL) and dried at 80 °C. Once dried, the Phy@Fe_3_O_4_ nanoparticles were milled to a fine powder and stored in a closed flask. The Phy@Fe_3_O_4_ nanoparticles were stable for at least three months (Figure 1).

## 3. Results

### 3.1. Characterization of Magnetite Nanoparticles

The co-precipitation method is a simple, fast, and affordable way to synthesize magnetic iron oxide nanoparticles (Fe_3_O_4_NPs) [30]. The synthesis procedure is based on the simultaneous precipitation of iron (II) and iron (III) oxides from the solution of the corresponding chlorides in an aqueous medium. The process is carried out under agitation and it uses a strong base (NaOH) to provide the mixed iron oxide, magnetite (Fe_3_O_4_), as nanoparticles, with a size of approximately 40 nm. The stoichiometric reaction is presented below.
2FeCl_3_ + FeCl_2_·4H_2_O + 8NaOH → Fe_3_O_4_ + 8NaCl + 8H_2_O

The synthesized pristine magnetite nanoparticles were decanted by an external magnetic field (NdBFe neodymium). After washing with abundant Milli-Q water to remove excess NaOH from the medium, the Fe_3_O_4_NPs were rinsed with MeOH to remove excess water, which could speed up the oxidation process. The nanoparticles were preserved by storing in methanol. Iron oxide nanoparticles were characterized by infrared spectroscopy (FT-IR), thermal analysis (TGA), transmission electron microscopy (TEM), dynamic light scattering (DLS), and zeta potential, consistent with the literature [31]. The Fe_3_O_4_NPs showed a pseudo-spherical shape with an average size of 41 nm. The TEM (Appendix A) displayed that the distribution of the MNPs was between 32.5 and 50 nm.

The DLS data in Appendix A was performed at pH 6.5 and it showed that the average hydrodynamic radius of the nanoparticles had a value of 288 nm with a polydispersity index (PdI) of 0.281. A small percentage of magnetite nanoparticles (3.6%) were aggregated with an average value of 4677 nm.

The surface of the magnetite nanoparticles is coated with hydroxyl groups as seen in FT-IR, which impacts the zeta potential. The zeta potential is also dependent on the pH of the environment (Appendix A). At pH 6.5 and 25 °C, the zeta potential was −21.3 mV.

The FT-IR spectrum of the magnetite nanoparticles is shown in Appendix A. It has the most relevant bands (3385 cm^−1^ and 1626 cm^−1^) corresponding to the OH groups on the surface of the nanoparticle and the residual water of the sample. The characteristic bands of the Fe–O–Fe bond are clearly observed at 576 cm^−1^.

FT-IR data suggest that the pristine surface of the Fe_3_O_4_NPs is mainly coated with hydroxyl groups (–OH). The hydroxyls can form hydrogen bonds with water molecules, as confirmed by the thermogravimetric analysis (TGA) of magnetite (Appendix A). Between 20 °C and 200 °C, there is a weight loss of 6.893% due to surface-bound water. Above 200 °C, there is a less significant weight loss (2.661%) due to residual water inside nanoparticles’ pores.

### 3.2. Characterization of Magnetite Nanoparticles Functionalized by Phytate Groups

The Phy@Fe_3_O_4_ NPs were prepared in a single three-minute step via vigorous stirring of all its components in basic medium. The new material had a black ferrofluid appearance that was easily precipitated by adding a few mg of NaCl to the solution, thus increasing the ionic strength of the medium. The Phy@Fe_3_O_4_ are stable once dry because they are covered by phytate groups, which prevents leaching. They can be stored for three months without any changes to their properties.

The coating of magnetite by phytate is revealed by comparing the FT-IR spectra of Fe_3_O_4_ NPs (blue curves), phytate salt (green curves), and Phy@Fe_3_O_4_ (red curves); see Figure 2. The new material presents characteristic bands of magnetite nanoparticles including a Fe–O–Fe bond-stretching band at 578 cm^−^^1^. The small displacement observed (from 576 and 578 cm^−^^1^) suggests that there is little change in the internal structure (core) of the magnetite. The broad band at 3385 cm^−^^1^ corresponds to the O–H tension bond and to water molecules that cover the surface of the nanoparticles. The signals at 1181, 1045, and 896 cm^−^^1^ indicate free phytate. Phy@Fe_3_O_4_ has displacement at 1120, 1057, and 890 cm^−^^1^, which can be assigned to P–O–Fe and the P–O stretching vibrations. The signal at 890 cm^−^^1^ is consistent with the P–O–H band disappearing in the Phy@Fe_3_O_4_ spectra. There is predominantly monodentate/bidentate binding of phytate to the surfaces of Fe_3_O_4_NPs [5].

TEM images of the Phy@Fe_3_O_4_ (Figure 3) show a pseudo-spherical morphology with and an average size of 110 nm. The size is not homogeneous but it varies between 80 to 160 nm. The average area value is 38,013 nm^2^.

Hydrodynamic diameter analysis by DLS was only recorded at pH = 6.09 because this is the approximate pH of the cation samples in the capture experiments. A size distribution between 50 nm and 800 nm was observed, with the average being approximately 196 nm. There is also a small population of fully aggregated particles over 4500 nm.

TGA plots of the Phy@Fe_3_O_4_ show a multi-stage mass loss (Figure 4a). The first interval between 80–120 °C can be considered as the loss of water molecules that are weakly bound to the surface of the nanoparticles, which represents 6.17%. The second interval near 200 °C, as well as the third between 270–300 °C, can be considered as the loss of mass corresponding to water molecules strongly bound on the surface of the nanoparticles or associated with hydrogen bonding to phosphate groups. This peak corresponds to a mass loss of 13.30%. Above 300 °C, a mass loss can be seen due to the organic fraction (myo-inositol) of Phy@Fe_3_O_4_. This represents a 4.06% loss and corresponds to the peaks marked by the derivatives located at 436.8 and 473.1 °C. Approximately 30% of phytate, i.e., the fraction of myo-inositol, is consumed by combustion. Thus, it can be assumed that the actual loss of phytate on Phy@Fe_3_O_4_ is 26%.

Phytic acid is a molecule with two different pKa values for the 12 acidic protons present. The charge value on the surface of the nanoparticle is highly dependent on the pH of the environment. Figure 4b shows the variation of zeta potential at different pH values in the medium. The zeta potential is always negative from pH 3 to 12; it is near −40 mV above pH 12. From pH 6.8 to 7.2, the zeta potential is around −35 mV suggesting high stability of the suspension of Phy@Fe_3_O_4_ nanoparticles. This negative value of the zeta potential suggests that the phosphate groups of the phytate remain at pH 7 and are completely deprotonated.

From the TGA data (mass loss 26%) and TEM (mean radius 110 nm), the number of phytate molecules per nm^2^ coordinated with the nanoparticle surface can be determined. Calculations show that the number of phytate molecules per nm^2^ (insertion coefficient) is 5.6. This insertion coefficient is an indicative measure that suggests that the functionalization of Fe_3_O_4_ nanoparticles is moderate [32].

### 3.3. Magnesium, Calcium, and Strontium Adsorption on Phytate-Fe_3_O_4_ NPs

Strontium is an alkaline earth metal that shares chemical-physical similarities with calcium and magnesium. Thus, we studied the retentive capacity of Phy@Fe_3_O_4_ nanoparticles for Ca(II), Sr(II), and Y(III) cations in non-competitive assays, as well as in competitive assays with all cations present in the same aqueous solution. In non-competitive assays, retention studies by Phy@Fe_3_O_4_ nanoparticles were performed separately for each cation. A 10 ppm solution of the cation at pH 7 was treated with 0.5 mg/mL suspended nanoparticles. The resulting suspension was stirred using a vibrating shaker at room temperature for two hours and decanted with a neodymium boron magnet. The remaining concentration of each cation was measured by ICP-OES. The competitive test was performed similarly with 10 ppm of Ca(II), Sr(II), or Y(III) in the initial solution.

In the non-competitive tests (Figure 5), a remarkable retention of the Sr(II) (75%) and Ca(II) (79%) ions on Phy@Fe_3_O_4_ nanoparticles can be seen. Instead, the Phy@Fe_3_O_4_ nanoparticles only show moderate retention of Mg(II) ion (37%).

Competitive tests (Figure 5) show a clear decrease of the retention of all cations by the phytate moiety. While magnesium capture drops to 3%, the retention percentages for Ca(II) (38%) and Sr(II) (47%) are moderate.

Phosphoric groups in its anionic form establishes strong 1:1 stoichiometry complexes with Mg(II), Ca(II), and Sr(II). Mg(II) has a very stable and totally soluble neutral form at neutral pH (due to its lower ionic radius); thus, it responds to a stoichiometry [Mg_5_(H_2_L)] where L represents the phytate molecule completely deprotonated. Ca(II) and Sr(II) have similar species at physiological pH, but they appears in very low concentrations. The solubility products for Ca(II) (pK_s_ = 39.3), Sr(II) (pK_s_ = 35.6), and Mg(II) (pK_s_ = 32.9), indicates that the solubility of the corresponding phytates increases as Mg > Sr > Ca. This sequence agrees with the retention percentages observed in this study [23].

In a complementary study (Appendix A), it can be observed that keeping the amount of phytate nanoparticles constant (2.5 mg/L) and varying the concentration of Sr(II) in the medium from 1 to 20 ppm has no significantly different values of retention, approximately 80%. These data indicate that the equilibrium point has been reached and is marked by the distribution constant.

In parallel, it is proved that the amount of Sr(II) retained depends on the amount of Phy@Fe_3_O_4_ nanoparticles used. In Appendix A, this variation is shown by studying 10 ppm of Sr(II) with variable amounts of Phy@Fe_3_O_4_ nanoparticles (5.1 mg to 0.32 mg). It is observed that the variation decreases exponentially.

### 3.4. Strontium and Yttrium Adsorption on Phytate-Fe_3_O_4_ NPs

It is known that when ^90^Sr is released into the environment after an accident, it rapidly decomposes into ^90^Y. Therefore, the retention capacity of Phy@Fe_3_O_4_ nanoparticles for both Sr(II) and Y(III) cations was studied. The analysis of non-radioactive solutions used an aqueous solution of 10 ppm of each cation at pH 7. Next, 2.5 mg of nanoparticles were introduced into the solution. The suspension was stirred for two hours and collected with a neodymium boron magnet. To fully promote the settling process, the ionic strength of the medium was increased with 50 μL of brine. The remaining concentration of each cation was measured using the ICP-OES. The results obtained shown a retention percentage of 75% for Sr(II) and 100% for Y(III).

The retention capacity of Phy@Fe_3_O_4_ nanoparticles was also analyzed with a sample of radioactive strontium. Radioactive strontium contains ^90^Sr and ^90^Y in equilibrium. Phy@Fe_3_O_4_ nanoparticles can separately recognize both metal cations, and the capture of each radioactive cations was determined next: 2.5 mg of Phy@Fe_3_O_4_ nanoparticles were added to 5 mL of a standard radioactive solution of 29.7 Bq at pH 6.5. This suspension was stirred using a vibrating shaker at room temperature for 2 h. The Phy@Fe_3_O_4_ nanoparticles were then collected with a neodymium-boron magnet. 3 mL of the supernatant solution were removed with a pipette and dried on an aluminum plate at 140 °C. The residue was evaluated with radiometric analysis.

The Phy@Fe_3_O_4_ nanoparticles were next dispersed in 2 mL of the solution, and the suspension was filtered with a 0.20-micron PVDF filter to recover the Phy@Fe_3_O_4_ nanoparticles. The samples radioactivity was recorded once they were dried on an aluminum plate at 140 °C. This procedure was repeated four times with 5 mL of radioactive solution (Figure 6). The coincidence of results for the uptake of Sr(II) using non-radioactive and radioactive samples is excellent within the experimental error, i.e., 73%, Table 1.

### 3.5. Uranium Adsorption on Fe_3_O_4_ NPs and Phytate-Fe_3_O_4_ NPs

The isoelectric point of magnetite nanoparticles is in the range of 6.8 to 7.2. At this pH, the uranium species U(VI) that exists in the medium are cationic: UO_2_^2+^, UO_2_OH^+^, and (UO_2_)_3_(OH_5_)^+^ [33]. Thus, the sorption of cationic uranium species on amphoteric groups of the magnetite is a favorable process.

As can be seen in Figure 7, the pristine Fe_3_O_4_NPs show a high uranium uptake activity of almost 100%. This is because Fe_3_O_4_NPs uptake uranium via adsorption over its negatively charged surface. Then, a redox process then occurs between the Fe(II) to Fe(III) of Fe_3_O_4_ NPs and uranium atoms U(VI)/U(IV) [34].

The pristine Fe_3_O_4_ NPs also show a moderate activity in the uptake of Y(III) (52%) and, to a lesser extent, with the Sr(II) ion; only 12%. There is no doubt that despite showing excellent uranium uptake, the stability of pristine magnetite is precarious because of the oxidation of Fe(II) to Fe(III) that occurs on the surface of the nanoparticle without an adequate coating. This is detrimental to the process.

On the other hand, the Phy@Fe_3_O_4_ NPs show an excellent capacity for the capture of radioactive ions: uranium (VI) (97%), yttrium (III) 100%, and strontium (II) of 78%, in the competitive tests studied. This test was performed by simultaneously adding 5 mL of 1000 ppb of each metal salt to 2.5 mL of 5 mg/L of Phy@Fe_3_O_4_ NPs. This mixture was stirred for two hours at room temperature and pH 6.5. The Phy@Fe_3_O_4_ NPs were then decanted using the NdBFe magnet. 1 mL of the supernatant solution was diluted in 10 mL of HNO_3_ (2%) and analyzed for salt content using ICP-OES.

Moreover, an ICP-OES adsorption study was performed to evaluate the maximum sorption capacity of the Phy@Fe_3_O_4_ NPs as a function of contact time. We analyzed the remaining supernatant concentration of U(VI) at different times (0–200 min). The initial concentration of U(VI) was 20 mgL^−1^ in contact with 2.5 mg of Phy@Fe_3_O_4_ NPs. Figure 8 shows that the maximum retention at equilibrium occurs after 2 h and remains constant at longer time points. This test was carried out at pH = 6.5 and 20 °C. The affinity of the Phy@Fe_3_O_4_ NPs for the uranyl cation is obvious: After one hour, only ~2 ppm of U(VI) remains in solution, which indicates that already 90% of U(VI) is adsorbed on the Phy@Fe_3_O_4_ NPs.

The variation in retention for a 20-ppm solution of uranyl ion with different quantities of Phy@Fe_3_O_4_ NPs nanoparticles was studied next: 0.25, 0.5, 1, 1.5, 2, and 2.5 mg Phy@Fe_3_O_4_ NPs were used. The capture percentages of U(VI) are all around 93% (Appendix A).

These tests used aerobic conditions without an inert atmosphere. Environmental carbon dioxide at pH ≥ 8 can be dissolved in the aqueous solution causing the formation of the complex [(CO_3_)_2_UO_2_]^4−^ that reduces the retention percentages [22]. Nevertheless, this material has practical value in real environmental conditions. The retention of 100% of U(VI) is equivalent to a q_m_ of 948 *±* 38 mg milligrams of U(VI) per milligram of Phy@Fe_3_O_4_ NPs used. This value is lower than others described in the literature [28] using magnetite nanoparticles with supported phosphate groups although the Phy@Fe_3_O_4_ nanoparticles presented here differ from the others by showing a high affinity for cations Y(III) and Sr(II) that did impact their uptake of U(VI) for the simultaneous presence of both radioactive ions.

The Phy@Fe_3_O_4_ NPs presented in this study are both effective and stable. When kept dry, they remain viable for more than three months with no loss of efficacy during subsequent uptake tests. Phosphonate coating causes a strong interaction with the surface of the magnetite and inhibit oxidation.

The TEM-EDS data of the decanted samples after use in the competitive capture of U(VI), Y(III), and Sr(II) are shown in Figure 9. In the orange box, the contamination due to the sample grid is highlighted. The green boxes show metals adsorbed on the surface of Phy@Fe_3_O_4_ NPs. The red arrow indicates phosphorus and iron from Phy@Fe_3_O_4_ NPs. These data confirm sorption effectiveness of Phy@Fe_3_O_4_ NPs for these radioactive metals.

## 4. Discussion

Many other papers have shown the recovery of U(VI), ^90^Sr, and ^90^Y salts. Table 2 shows a non-exhaustive list of adsorbent materials including removal or adsorption capacity, pH, and equilibrium time for optimal removal of U(VI), Sr(II), and Y(III). The results obtained here are also included.

Of the three cations studied here, the uranyl cation UO_2_^+^ has received the most attention from researchers. Therefore, different adsorbent materials of different origin have been described in the literature. Supramolecular receptors based on calixarenes [35,36,37] are particularly powerful as they are nanostructured porous materials, MOFs [38] recyclable chelating resins [39], polymeric chitosan functionalized with triethylene tetramine [40], porous polymers decorated with phytic acid [41], and some nanomaterials derived from iron oxide including magnetite functionalized with phosphate groups [34].

^90^Sr salt uptake has been described via crown ethers such as di-tert-butyldicyclohexano-18-crown-6 supported on magnetite or on maghemite [42,43,44]. Other systems include porous metal-organic frameworks (MOFs) [45,46,47], zeolites LTA (Linde Type A) [48], polymers of melanina [49], and derivatives of calixarenos (carboxy methoxy calix[6]arene [50] and thiacalix[4]arene diamide [51]).

The Y(III) salts can be eliminated from aqueous medium by precipitation with iron oxyhydroxide compounds [52]. Sorption on kaolinite [53], sodium alginate (99.01 mg/g) [54], or calcium alginate (181.81 mg/g) has been described. Nanomaghemite has been used effectively for removal of Y(III) salts (13.5 mg/g) [55].

The Phy@Fe_3_O_4_ NPs presented here offers a high adsorption capacity at equilibrium (q_m_). The iron oxide nanomaterial adsorbs more U(VI) from an aqueous solution relative to the literature. To the best of our knowledge, the Phy@Fe_3_O_4_ nanoparticles are the only receptors based on iron oxide described in the literature that can eliminate salts of U(VI), Y(III), and Sr(II) without interference.

## 5. Conclusions

In this work, a new hybrid material has been prepared and characterized. Phy@Fe_3_O_4_ NPs has been obtained via conjunction of magnetite nanoparticles supported with phytate moieties. The new hybrid material is stable, affordable, and quick to prepare (3 min). The Phy@Fe_3_O_4_ NPs show a high sorption capacity of the radioactive ions studied under real aerobic conditions. The system eliminates uranyl ions (100%), Y(III) (100%), and Sr(II) (73%) at pH 6.5, which is typical for most environmental samples.

Pristine magnetite nanoparticles, Fe_3_O_4_ NPs, have a high surface reactivity and also show a quantitative sorption for U(VI), but much lower for Sr(II) (12%) and Y(III) (52%). Without post-coating, pristine magnetite nanoparticles are not stable, and a co-precipitation preparation method requires more synthetic effort and longer reaction times.

Phy@Fe_3_O_4_ NPs are non-toxic and easily recovered via their magnetic characteristics. They are suitable for capturing dangerous radioactive substances. Due to their simplicity and versatility, the Phy@Fe_3_O_4_ NPs have value in the circular economy. They can eliminate and confine dangerous radioactive metal ions such as ^90^Sr, ^90^Y, and (UO_2_)^2+^ from water.

## Figures and Tables

**Figure 1 nanomaterials-12-04383-f001:**
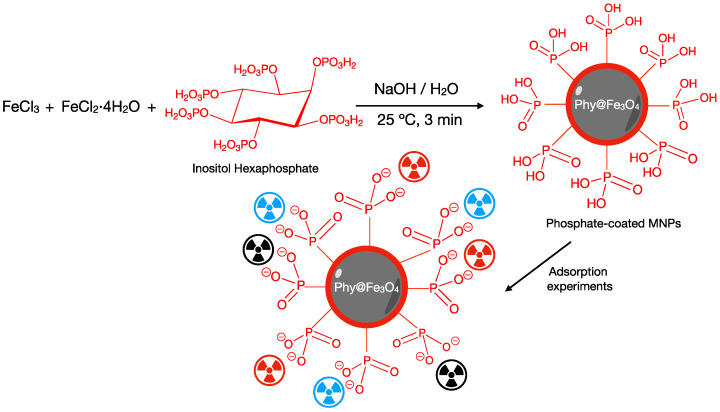
Scheme of the procedure of synthesis and functionalization of the magnetite nanoparticles with phytate. Schematic illustration on the adsorption mechanism.

**Figure 2 nanomaterials-12-04383-f002:**
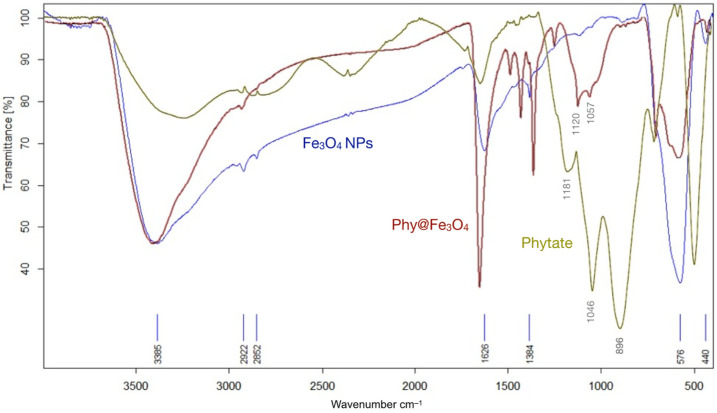
Infrared spectra (FT-IR) of different nanomaterials: Magnetite nanoparticles (Fe_3_O_4_NPs); Phytic acid sodium salt hydrate (Phytate); and magnetite nanoparticles covered with phytate groups (Phy@Fe_3_O_4_).

**Figure 3 nanomaterials-12-04383-f003:**
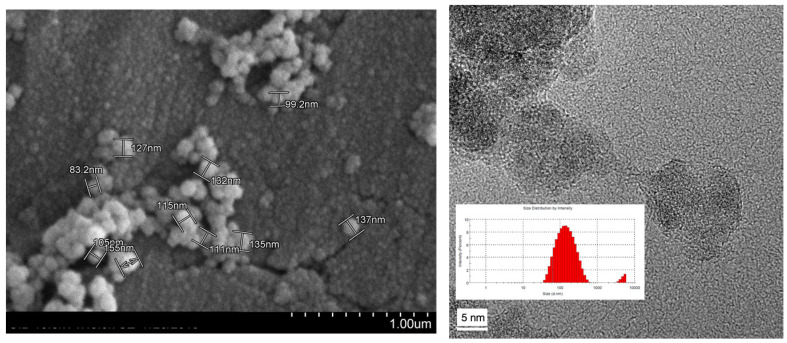
TEM micrographs and histogram showing the morphology and particle size distribution of magnetic nanoparticles covered by phytate groups (Phy@Fe_3_O_4_).

**Figure 4 nanomaterials-12-04383-f004:**
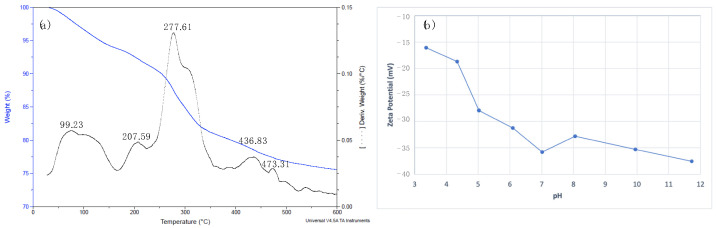
(**a**) TGA diagram of Phy@Fe_3_O_4_ nanoparticles under nitrogen atmosphere. (**b**) z-potential obtained for the Phy@Fe_3_O_4_ nanoparticles at different pH.

**Figure 5 nanomaterials-12-04383-f005:**
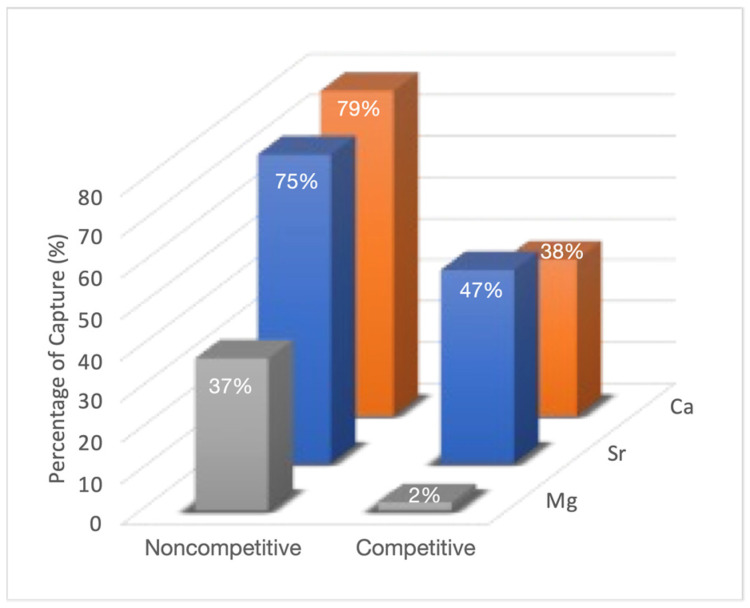
ICP-OES percentage retention results obtained by the analysis of 0.5 mg/L of Phy@Fe_3_O_4_ nanoparticles with a 10 ppm of Mg(II), Sr (II) and Ca(II) solution in noncompetitive and competitive media.

**Figure 6 nanomaterials-12-04383-f006:**
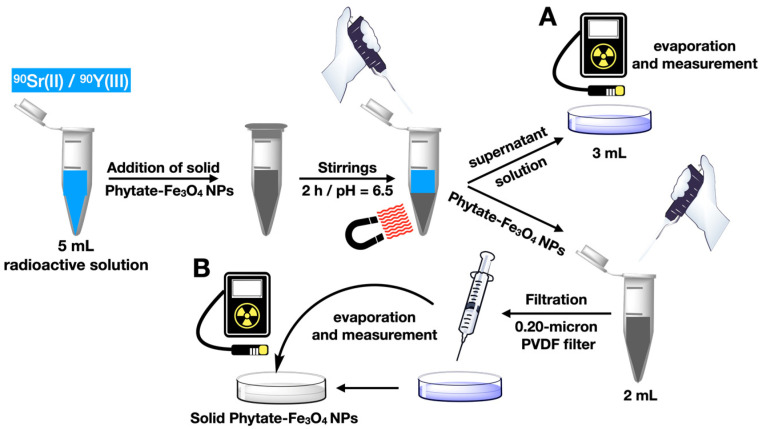
Scheme of the methodology followed for the recovery of radioactive strontium and yttrium from a standard sample.

**Figure 7 nanomaterials-12-04383-f007:**
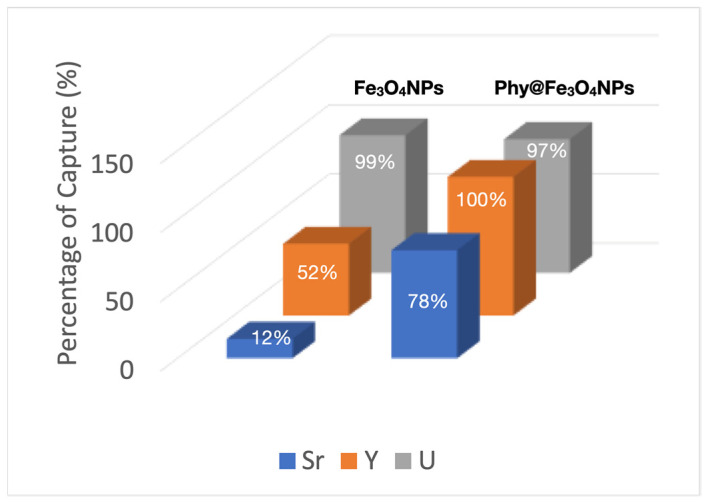
Capture percentages obtained for Sr(II), Y(III) and UO_2_^2+^ in a competitive test, using Fe_3_O_4_ NPs and Phy@Fe_3_O_4_ NPs.

**Figure 8 nanomaterials-12-04383-f008:**
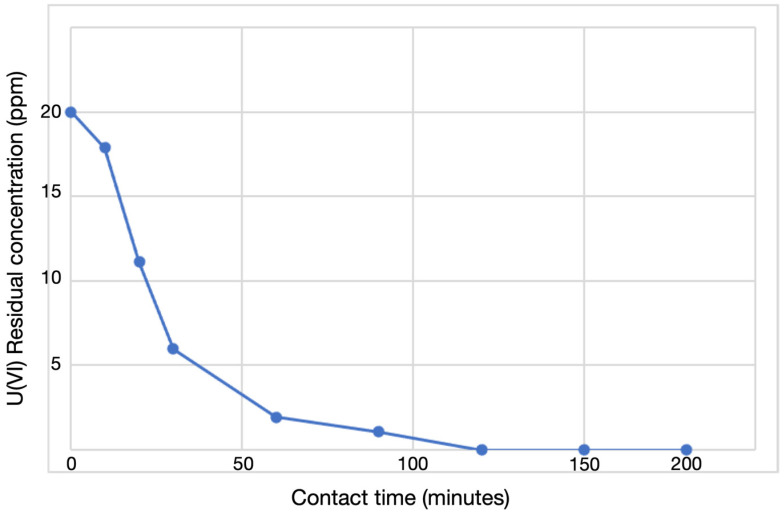
ICP-OES data of the supernatants collected after U(VI) adsorption on Phy@Fe_3_O_4_ NPs.

**Figure 9 nanomaterials-12-04383-f009:**
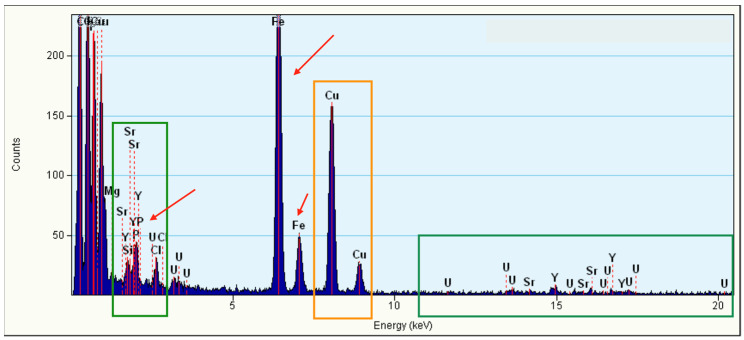
EDS spectrum acquired in TEM mode for a Phy@Fe_3_O_4_NPs sample. The orange boxes highlight the contributions from contamination on the sample grid, whereas the green box highlights the signal generated by the phosphate coating.

**Table 1 nanomaterials-12-04383-t001:** Results obtained by applying the ^90^Sr (II) and ^90^Y(III) uptake methodology using Phy@Fe_3_O_4_ nanoparticles.

Test	Initial Radioactivity Bq ^1^	% Radioactivity Measured in A ^2^	% Radioactivity Measured in B ^2^	% Total
1	73.43	31	68	99
2	72.75	31	71	102
3	74.00	32	70	102
4	72.19	30	74	104
Average	73.09 ± 0.79	31 ± 0.82	70.75 ± 2.5	101.75 ± 2.06

^1^ Bq = Becquerel; ^2^ See Figure 6.

**Table 2 nanomaterials-12-04383-t002:** Comparison of U(VI), Sr(II), and Y(III) adsorption capacities reported for adsorbents in published studies and the material in this work.

Sorbent	Adsorption or Removal Performance	pH/Time	Ref.
**URANIUM**			
Biorecovery of uranium from aqueous solutions at the expense of phytic acid	q_m_ = 4.1 mg/g 75%	4.5/12 h	[56]
Fe_3_O_4_ NPs (50–100 nm)	q_m_ = 5 mg/g	7/5 h	[34]
Phytic acid-decorated porous organic polymer for uranium extraction under highly acidic conditions	q_m_ = 106.7 mg/g	5/11 h	[41]
Iminodiphosphonate functionalized PGMA–Fe_3_O_4_ ^1^	q_m_ = 147 mg/g	4–5/12 h	[57]
Triethylene tetramine functionalized chitosan resin–Fe_3_O_4_	q_m_ = 166.6 mg/g	5/1 h	[40]
Amidoxime functionalized flower-like TiO_2_ microspheres–Fe_3_O_4_	q_m_ = 313.6 mg/g	6/12 h	[58]
Graphene oxide modified with OPO_3_H_2_/mesoporous Zr-MOF–Fe_3_O_4_	q_m_ = 416.7 mg/g	6.2/3 min	[59]
Diethylenetriamine Fe_3_O_4_-embedded mesoporous silica	q_m_ = 470 mg/g	6/5 h	[60]
Phytic acid/polyaniline/FeOOH composites	q_m_ = 555.8 mg/g (Removal efficiency = 92%)	8/5 min	[61]
Oleic acid bilayer@Fe_3_O_4_ NPs	q_m_ = 635 mg/g	5.6/24 h	[62]
Hydrothermal carbon modified with NaOH–Fe_3_O_4_	q_m_ = 761.2 mg/g	5.5/200 min	[63]
Phytic acid doped polypyrrole/Carbon felt electrode	q_m_ = 1562 mg/g (Removal efficiency = 98%)	5/8 h	[64]
Oleyl phosphate bilayer@MnFe_2_O_4_ NPs	q_m_ = 1667 mg/g	5.6/24 h	[65]
Phosphated-Fe_3_O_4_ NPs	q_m_ = 1690 mg/g (100%)	7/1 min	[24]
Phy@Fe_3_O_4_NPs	q_m_ = 948 mg/g (Removal efficiency = 80%)	6.5/2 h	This work
**STRONTIUM**			
4′,4″(5″)-di-tert-butyldicyclohexano-18-crown-6 onto Fe_3_O_4_@UiO-66-NH_2_	q_m_ = 284 mL/g	1/90 min	[43]
γ-Fe_2_O_3_ impregnated with 4′,4″(5″)-di-tert-butyldicyclohexane 18-crown-6	(Removal efficiency = 66%)	7/10 min	[42]
Dicyclohexano-18-crown-6 ether impregnated titanate nanotubes	q_m_ = 49 mg/g	1/180 min	[66]
MOF material	q_m_ = 43.83 mg/g (Removal efficiency = 99,89%)	7/12 h	[47]
Ammonium molybdophosphate–polyacrylonitrile (AMP–PAN)	q_m_ = 15 mg/g (Removal efficiency = 66%)	5/24 h	[67]
Phy@Fe_3_O_4_NPs	q_m_ = 20 mg/g (Removal efficiency = 73%)	6.5/2 h	This work
**YTTRIUM**			
Nanomaghemite	q_m_ = 13.5 mg/g (Removal efficiency = 94%)	6.9/50 min	[55]
3-Amino-5-Hydroxypyrazole Impregnated Bleaching Clay	q_m_ = 171.3 mg/g	6/60 min	[68]
Alginate compounds	q_m_ = 99.01 mg/g (sodium alginate) q_m_ = 181.81 mg/g (calcium alginate)	6/60 min	[54]
Kaolinite	q_m_ = 0.029 mmol/g	5/3.5 h	[53]
Phy@Fe_3_O4NPs	q_m_ = 20 mg/g (Removal efficiency = 100%)	6.5/2 h	This work

^1^ PGMA: polyglycidyl methacrylate-magnetic nanocomposite.

## Data Availability

Not applicable.

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
