# Peer review of "One-Pot Environmentally Friendly Synthesis of Nanomaterials Based on Phytate-Coated Fe3O4 Nanoparticles for Efficient Removal of the Radioactive Metal Ions 90Sr, 90Y and (UO2)2+ from Water"

_nanomaterials, 2022, doi:10.3390/nano12244383_

Round 1

Reviewer 1 Report

I think the present study is very interesting and would benefit greatly related removal researches. Several suggestion is for reference.

1. In Title, suggest 90Y and (UO2)2+”.

2. In Keywords, suggest adsorption removal” and radioactive metal ions.

3. Variables should be italicized, and the invariant shouldnt.

4. Please add a schematic illustration on the adsorption mechanism for easily understanding.

5. The two related papers can be cited for reference,

-One-pot synthesis of magnetic iron oxide nanoparticle-multiwalled carbon nanotube composites for enhanced removal of Cr(VI) from aqueous solution, Journal of Colloid and Interface Science 505 (2017) 1134-1146.

-One-pot synthesis of magnetic molecularly imprinted microspheres by RAFT precipitation polymerization for the fast and selective removal of 17β-estradiol, RSC Adv., 2015, 5, 10611-10618.

6. The total manuscript needs language optimization.

Reviewer 2 Report

The problem of the radioactive contamination of water is as old as the nucleonic sciences, and the problem of purification of water from poison radionuclides has been known for as many years as its harmful effects on the human body. So, any work on this topic is important. The peer-reviewed publication also. Therefore, I began reading it with great interest, as I was intrigued by the abstract declaring a time of three minutes sufficient for the synthesis of the sorbent.

However, as I read more and more, I became convinced that the article requires major revisions.

Below, are just a few comments:

Title/abstract: 

(1) I do not know why exactly these radionuclides were chosen. The authors don't explain it anywhere in the text.

(2) Strontium-90 remains in equilibrium with Y-90 and even after purification  (e.g. on the AG 50W/X8 column) after about 30 days it comes back to an equilibrium with yttrium. So, there is no need to sorb these radionuclides separately. I expect, that in the novel version, the authors will discuss this problem.

(3) Making a cursory, even, review of the existing literature, it can be concluded that there are equally fast methods for purifying water from radionuclides. Some of them use even magnetic sorbents. For example, the one-pot synthesis/sorption using the polyvalent metal alginates [Fuks, L., Herdzik-Koniecko, I., Polkowska-Motrenko, H., Oszczak, A., Novel procedure for removal of the radioactive metals from aqueous wastes by the magnetic calcium alginate, (2018) Intern. J. Environm. Sci. Technol., 15, 2657-2668].

Lines 323 and the following:

Ionic radii of Mg2+, Ca2+ and Sr2+ are 0.065pm, 0.099p, and 0.113pm, respectively. I have not found even one sentence of explanation, as to why the solubility of the phytates forms a sequence Mg>Sr>Ca. Even in the previous papers of the same authors (e.g. [23]).  I can not find the discussion of the phenomenon. By the way, the ionic radius does not determine the strength of the ionic interaction, but the ionic potential (i.e. Z/r).

Line 321: Phosphoric groups of the phytate form the complexes with metals, not the whole molecule of the ligand.

Table 1: %Total, as an experimental value, does not exceed 100%. Please give the SD values.

Table 2: the data shown by the authors do not support the thesis about the superiority of the proposed sorbents if only the adsorption removal performance is analysed.

Figures 5 and 8 are not mentioned in the text.

Also, taking into account the found typos and grammatical errors, the text presented to me requires further re-thinking by the authors and changing the description of the discussion and conclusions. As a result, I suggest that the work presented to me requires major revisions.
